# Tree-Ring Based Chronology of Landslides in the Shirakami Mountains, Japan

Kinuko Noguchi [1,2], Ching-Ying Tsou [1,*], Yukio Ishikawa [3], Daisuke Higaki [4] and Chun-Yi Wu [5]

1   Faculty of Agriculture and Life Science, Hirosaki University, Hirosaki 036-8561, Japan; noguchi.kinuko.k0@elms.hokudai.ac.jp
2   Graduate School of Agriculture, Hokkaido University, Sapporo 060-8589, Japan
3   Shirakami Research Center for Environmental Sciences, Faculty of Agriculture and Life Science, Hirosaki University, Hirosaki 036-8561, Japan; yishi@hirosaki-u.ac.jp
4   Nippon Koei Co., Ltd., Tokyo 102-8539, Japan; a9024@n-koei.co.jp
5   Department of Soil and Water Conservation, National Chung Hsing University, Taichung 40227, Taiwan; cywu@nchu.edu.tw
*   Correspondence: tsou.chingying@hirosaki-u.ac.jp

**Abstract:** The N-Ohkawa landslide, and the southern section of the Ohkawa landslide, occurred during the snow-melt seasons of 1999 and 2006, respectively, in the Shirakami Mountains, Japan. This paper examines the response of trees in the Shirakami Mountains to landslides, and also investigates the spatio-temporal occurrence patterns of landslide events in the area. Dendrogeomorphological analysis was used to identify growth suppression and growth increase (GD) markers in tilted deciduous broadleaved trees and also to reveal the timing of the establishment of shade-intolerant tree species. Analysis of the GD markers detected in tree-ring width series revealed confirmatory evidence of landslide events that occurred in 1999 and 2006 and were observed by eyewitnesses, as well as signals from eight additional (previously unrecorded) landslide events during 1986–2005. Furthermore, shade-intolerant species were found to have become established on the N-Ohkawa and southern Ohkawa landslides, but with a lag of up to seven years following the landslide events causing the canopy opening.

**Keywords:** tree ring; dendrogeomorphology; landslide; landslide activity; deciduous broadleaved tree; Shirakami Mountains

## 1. Introduction

Landslides are common in mountainous regions, and can be driven by tectonic, climatic, and/or human activities [1,2]. Landslides can create permanently unstable sites, and as a result, can drastically alter landscape morphology, damage forest environments, and even endanger life. Identifying the spatial and temporal patterns of landslide occurrence is vital for environmental management and minimizing the losses associated with landslides. However, information regarding past landslide events is scarce and almost always incomplete.

Dendrogeomorphology can be used as a proxy indicator of past landslide activity at the scale of years [3–5]. This dating technique is based on the analysis of annual growth rings in trees, with the mixed signals being filtered to isolate the signal indicative of landslide events from non-landslide disturbances, such as climate variations, insect epidemics, and human activity, encoded within the tree-ring chronologies [6,7]. Landslides cause disturbances in tree growth that are preserved as variations within the tree-ring width series. These growth disturbances (hereafter GD) can take several forms, namely, abrupt growth release (wider annual rings), suppression (narrower annual rings), and the formation of compression wood that results from the elimination of neighboring trees, damage to the root, crown or stems, and stem tilting [5,8]. Dating of landslide reactivation

by interpretation of these GD markers preserved within annual-ring-width series has been performed using a moving-window approach to smooth out non-landslide fluctuations [9] or evaluating the change rate of the annual ring width if it exceeds a certain threshold value [10]. Additionally, other studies have dated landslides using different thresholds (e.g., the event-response ($I_t$) index and number of GD markers) [10,11]. Although the amount of research has increased in recent years, no systematic standard approach has yet been proposed and the choice of an appropriate definition and threshold appears to be site-specific.

Dendrogeomorphological studies of landslides have been performed using conifers in the European Alps and Americas [5,8,12]. In North America, Carrara [13] identified synchronous abrupt reductions in annual ring width in tree samples. He suggested that these tree responses were the result of damage during a landslide and was thus able to date the landslide event to 1693 or 1694 and infer that the trigger was an earthquake. With a focus on abrupt reductions in annual ring width and the formation of compression wood on the tilted side stem in the French Alps, Lopez-Saez et al. [10] assessed eight different stages of landslide reactivation over the past 130 years and found that landslide reactivation was associated with seasonal rainstorms. Recently, Lopez-Saez et al. [14] added abrupt increases in annual ring width as another type of growth disturbance, and this enabled reconstruction of 26 reactivation phases of landslides between 1859 and 2010 in the Swiss Alps. In the Orlické hory Mountains (Czech Republic), Šilhán [11] found that landslide activity is particularly associated with slide and creep effects, and the consequent growth disturbance can be identified in trees growing on the scarp and the landslide block. In contrast, there have been few such studies in Asia [3,12,15]. Recent studies have demonstrated that broadleaved trees are also useful for dating landslides and shown the need for additional case studies that consider, for example, an adequate variety of species and age classes [5,12].

Coherent landslides, which often move slowly (*Jisuberi* in Japanese), dominate in the Shirakami Mountains [16], but historical records relevant to landslide activity are scarce. In this study, we investigate the spatio-temporal patterns of landslide occurrence through analysis of the dendrogeomorphological record of 90 deciduous broadleaved trees from 12 species growing on landslide scarps and landslide moving bodies, which we refer to as the displaced blocks, on the right flank of the Ohakawa River, a tributary of the Iwaki River, within the Shirakami Mountains. Our main aims are: (i) to identify and interpret the GD markers (i.e., abrupt growth increase and growth suppression) preserved in the tree-ring series of trees growing on the landslide slopes; (ii) to investigate how these trees responded to landslides known to have occurred in the area; and (iii) to reconstruct the spatial and temporal patterns of landslide occurrence over the past 70 years using our GD data, as well as the timing of the establishment of shade-intolerant trees, and compare this with the limited eyewitness reports of landslides.

## 2. Study Area

The coherent landslides studied here were located on the right bank, and on an outside bend, of the meandering Ohkawa River, which originates from the eastern side of the Shirakami Mountains, northern Honshu Island, Japan (Figure 1). These landslides are covered by deciduous broadleaved trees dominated by Siebold's beech (*Fagus crenata*). The forest is a naturally regenerated, unmanaged secondary forest that developed after the original forest was selectively felled until 1967 [17]. The study area has a cool-temperate climate, with an average temperature of 8.1 °C and average annual rainfall of 2589 mm [18]. Each year, from November to the following April, the area is covered by snow to a maximum depth of about 2.2 m [18].

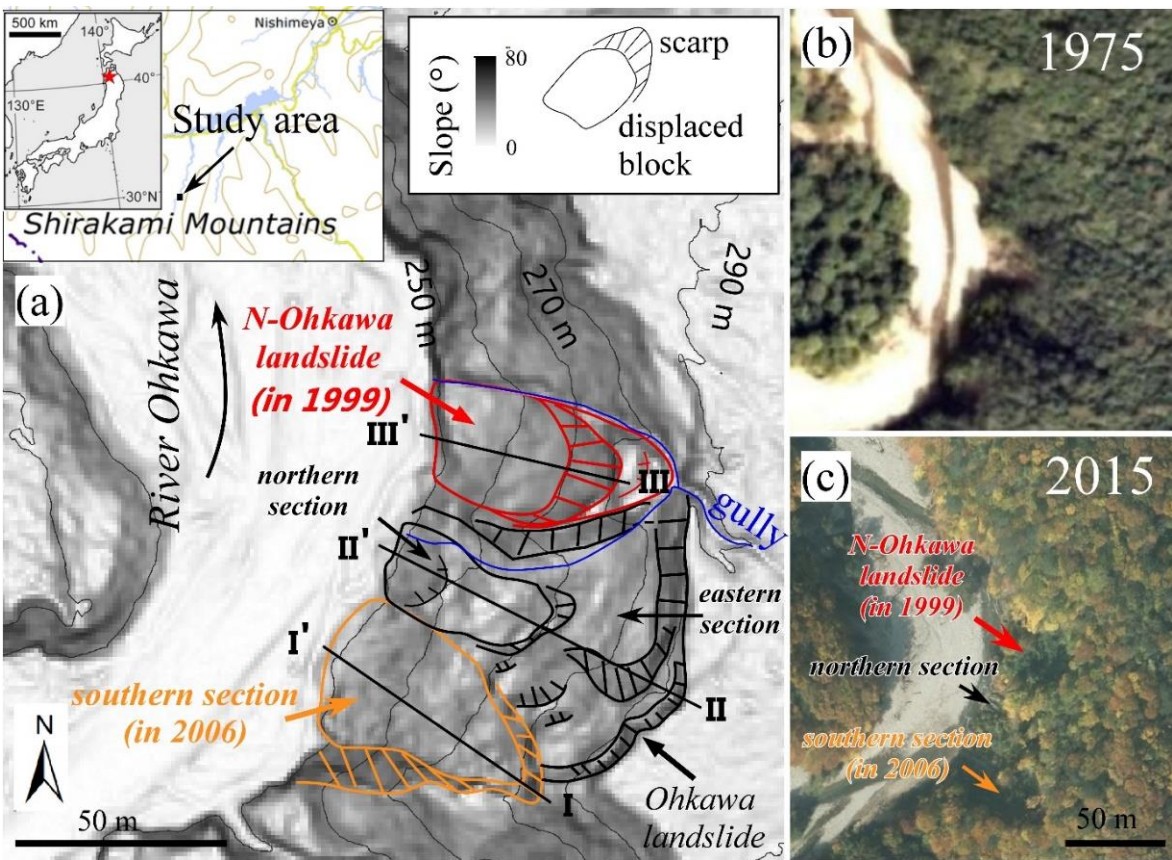

**Figure 1.** Coherent landslides, topographic map, and aerial photographs from the study area. (**a**) Landslide topography. Aerial photographs are from (**b**) 1975 and (**c**) 2015. The topographic map was constructed from a 1-m digital elevation model (DEM) based on LiDAR data provided by the Geospatial Information Authority of Japan (https://www.gsi.go.jp/, accessed on 12 April 2020). The landslide topography was interpreted using the slope image and the results were checked in the field. Topographic cross-sections (I–I', II–II', and III–III') are shown in Figure 2.

Our study area contains two neighboring landslide slopes: the N-Ohkawa and Ohkawa landslides, that are located along a 40-m-high terraced scarp, with the river terrace top at elevations of 285 to 295 m (Figures 1a and 2). Terrace gravels were exposed at the edge of the terrace after the landslides. The bedrock is formed from the mid-Miocene Hayaguchigawa Formation, which consists primarily of acidic pyroclastic deposits, but also contains andesitic pyroclastic deposits, sandstones, and conglomerates [19] (Figure 2). The N-Ohkawa landslide comprises a single displaced block. In contrast, distinctive stair-like features are evident on the displaced block of the Ohkawa landslide, which also comprises two secondary scarps that separate the individual blocks within the larger block at its northern and southern ends (Figure 2). Minor gully features are present in the landslide slope. Based on its slope geometry, we divided the Ohkawa landslide into three sections; i.e., the eastern, northern, and southern sections, for the following discussion. The timing of these movements is not well constrained. However, limited information obtained from several eyewitness accounts recorded during site visits suggests that the major movements of the N-Ohkawa landslide and the southern section of the Ohkawa landslide occurred in April 1999 and May 2006, respectively [20]; other slope movements of the Ohkawa landslide occurred recently, as described in Section 4.3. In addition, the lower slope of the N-Ohkawa landslide seems to have failed beforehand, as indicated by the bare area seen on the aerial photograph from 1975 (Figure 1b). The N-Ohkawa landslide and the northern and southern sections of the Ohkawa landslide are visible on the aerial photograph from 2015 (Figure 1c).

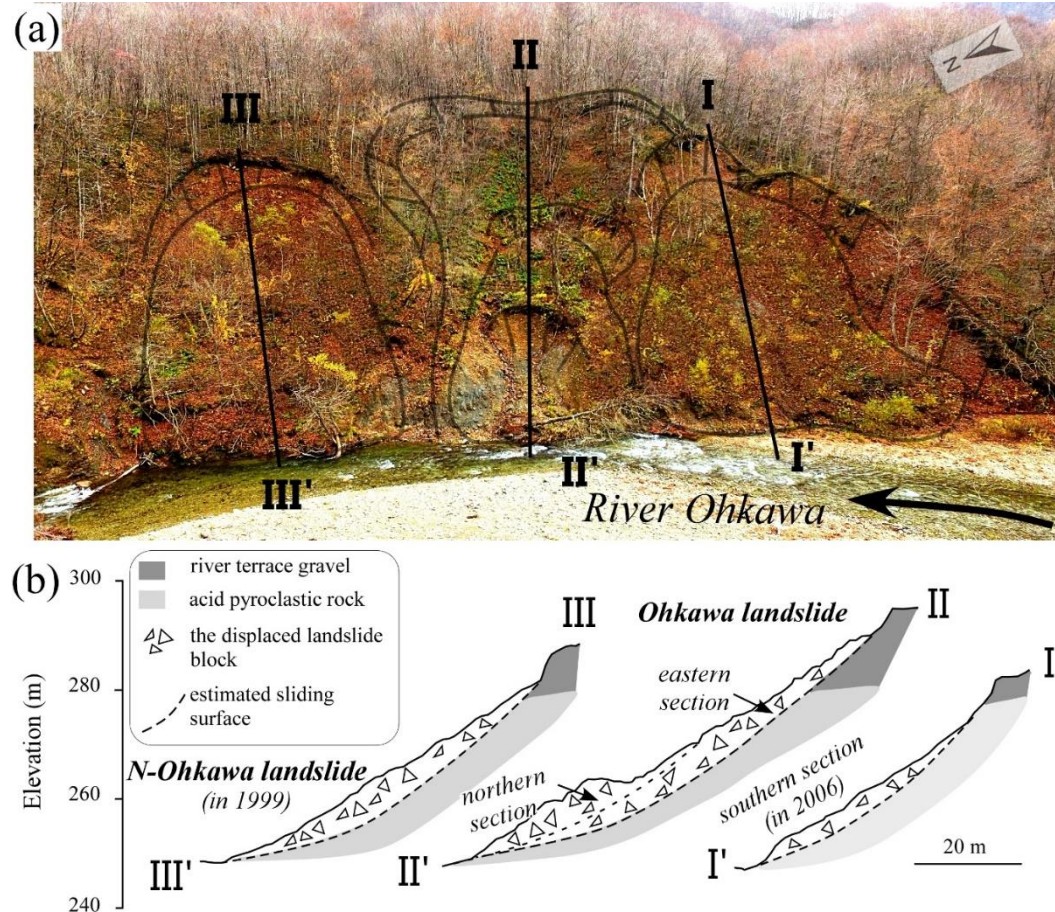

**Figure 2.** Topography and geological cross-sections of the studied landslide slopes. (**a**) A photograph of the study area. (**b**) Geological cross-sections of the N-Ohkawa landslide (III–III'), the eastern and northern sections of the Ohkawa landslide (II–II'), and the southern section of the Ohkawa landslide (I–I'). The photograph was taken in 2017. The cross-sections are based on the LiDAR DEM.

## 3. Methods

### 3.1. Sampling and Cross-Matching of Ring-Width Series

Increment cores were extracted from the upper side of the tilted stems of 90 living broadleaved trees using a Pressler increment borer (maximum length of 40 cm and diameter of 5.15 mm) between June and November 2019, on the main and secondary landslide scarps and on landslide-displaced blocks (Figure 3). The trees were sampled at trunk heights of 20–120 cm. According to the standard methods of dendrochronological research, increment core should be taken parallel to contour to avoid the development of reaction wood in tilted trees [21]. Tension wood develops on the upper side of leaning hardwood trees and typically has wider annual rings than on the lower side [3]. However, in the present study, we obtained cores oriented in the slope direction, because the formation of tension wood is, in itself, a good indicator of landslide movement [3]. Indeed, tension wood may not form in all tilted trees; therefore, wider annual rings may also be the result of growth release owing to, for example, the formation of canopy opening after landslide [5,8]. As such, responses resulting from both tilting and gap formation after landslides are included in our results.

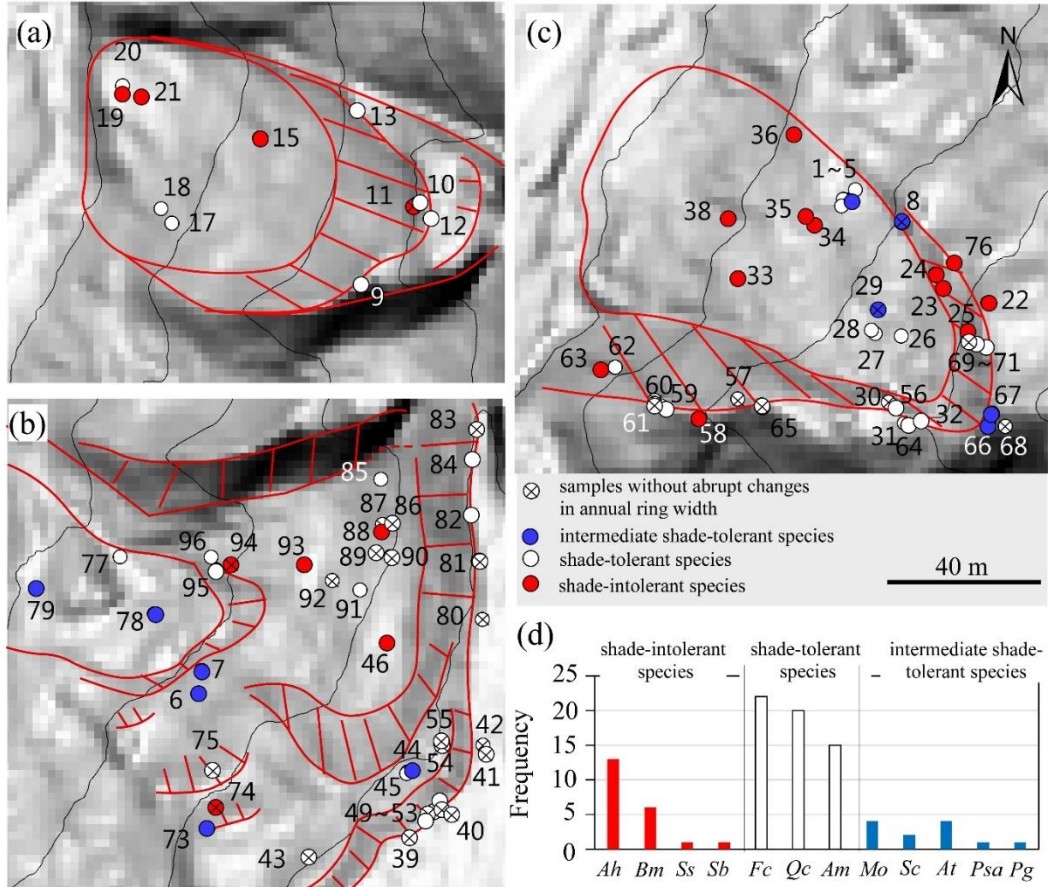

**Figure 3.** Locations of sampled trees and frequency distribution of tree species. (**a**) Locations of sampled trees at the N-Ohkawa landslide. (**b**) Locations of sampled trees in the eastern and northern sections of the Ohkawa landslide. (**c**) Locations of sampled trees in the southern section of the Ohkawa landslide. (**d**) Frequency distribution of tree species. The numbers on the maps are sample ID numbers.

We selected 21 samples from four shade-intolerant species (*Alnus hirsuta* (*Ah*), *Betula maximowicziana* (*Bm*), *Salix bakko* (*Sb*), and *Salix sachalinensis* (*Ss*)), 57 samples from three shade-tolerant species (*Acer pictum subsp. mono* (*Am*), *Fagus crenata* (*Fc*), and *Quercus crispula* (*Qc*)), and 12 samples from five intermediate shade-tolerant species (*Aesculus turbinata* (*At*), *Magnolia obovata* (*Mo*), *Prunus grayana* (*Pg*), *Prunus sargentii* (*Psa*), and *Sorbus commixta* (*Sc*); Figure 3). We collected 11 samples from the N-Ohkawa landslide and 79 samples (including 39 from the southern (2006) section) from the Ohkawa landslide. The cores were prepared and analyzed using standard procedures following Stokes and Smiley [22] and Speer [21]. The sample cores were prepared using a razor blade to maximize the visual resolution of the ring widths and were measured to the nearest 0.01 mm under a binocular zoom microscope (Olympus SZ61) using a precision measurement stage (Chuo Seiki LTD. LS-252D) attached to a digital output unit (Mitsutoyo Digimatic). After measurement, all cores were visually cross-dated by matching well-defined wide or narrow rings. In addition, longer chronologies (>60 years) of shade-tolerant species and several intermediate species were cross-dated by using a simple list method [23].

### 3.2. Identification of Growth Disturbance by Landslides in Tree-Ring Width Series and Age Determination of Shade-Intolerant Species

In this study, we considered two types of GD markers in the tree-ring width series: abrupt growth increase and abrupt growth suppression. GD markers were identified using the method described by Ishikawa et al. [7], in which a five year moving average of ring width is used to identify periods of abrupt growth increase or suppression as follows. A

growth increase is defined as a doubling of the five year moving average of the ring width when compared with that of the previous five year period and a defined growth rate that fluctuates continuously above 1 for at least 10 consecutive years. Conversely, a growth suppression is defined as a halving of the five year average ring-width and a growth rate fluctuating continuously below 1 for at least 10 consecutive years. In addition, because of spatial irregularities in tree growth, the duration of the GD also depends on the sampling position [4,13]. Therefore, we took into account moderate levels of GD in which the defined growth rate persisted for less than 10, but more than five, consecutive years. In some cases, there is a slight time lag from the casual disturbance event in the GD markers extracted using the moving average method because of growth variation prior to and/or after the event. To avoid this inaccuracy, we carefully checked the ring-width pattern around the timing of the GD markers, and used the information to decide on the GD marker years in the tree-ring width series. Figure 4a–d shows representative examples of how the GD marker years were identified in the tree-ring series using the above method. No significant changes in annual ring width were found in 30 of the cores from the sampled trees (33%) and these cores were not considered for further analysis.

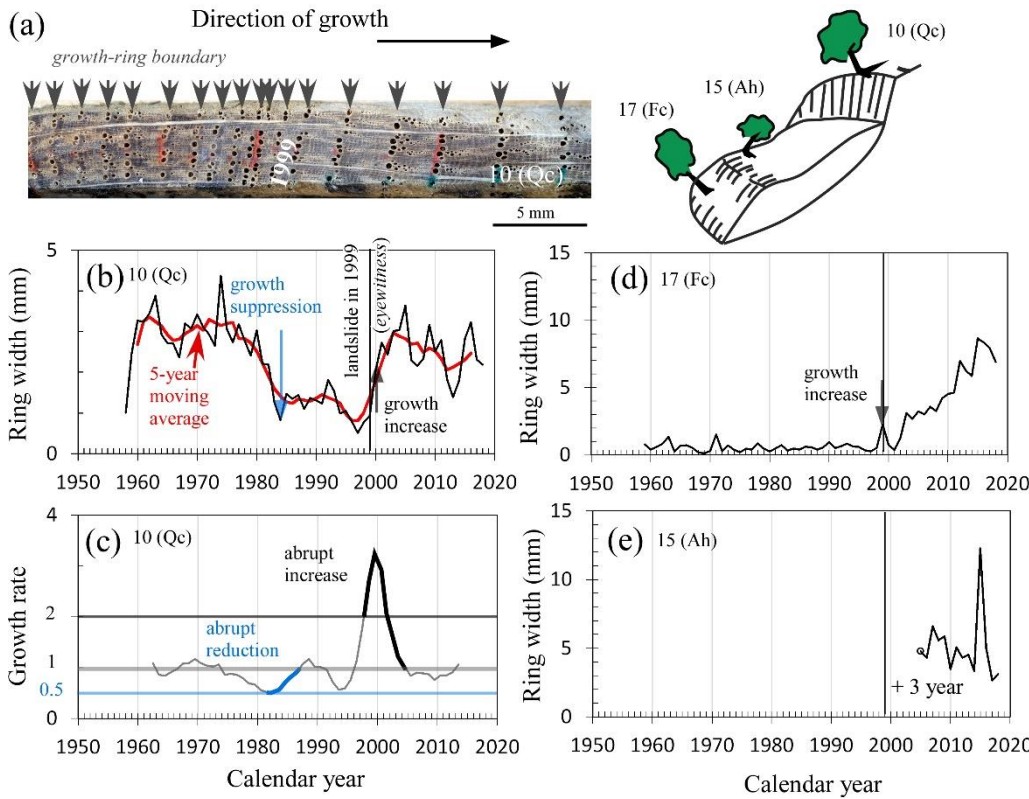

**Figure 4.** Representative cases from the N-Ohkawa landslide. (**a**) A micro-section of *Nu*. 10 (*Quercus crispula, Qc*) on the landslide scarp showing an abrupt increase in annual ring width in 1999. (**b**) A tree-ring width series and the 5 year moving average of the tree-ring width series of *Nu*. 10. Light blue and black arrows indicate identified response years of GD, growth suppression, and growth increase, respectively. (**c**) GD (i.e., growth suppression and growth increase) defined using growth rate of the tree *Nu*. 10. Note that the defined growth rate of growth suppression was continuously below 1 for 6 consecutive years, and that of growth increase was continuously above 1 for 7 consecutive years. (**d**) The tree-ring width series and identified GD from *Nu*. 17 (*Fagus crenata, Fc*) on the displaced landslide block. The annual ring width increased abruptly in 1999, followed by successive decreases in 2000 and 2001, and then by an increasing trend. (**e**) The tree-ring width series from *Nu*. 15 (*Alnus hirsuta, Ah*, the dominant shade-intolerant species in the study area). Open circle indicates the number of years for the tree to grow to the sample height.

The chronology of each of the previous landslides was expressed using the event-response ($I_t$) index, following Shroder [24], as follows:

$$I_t(\%) = \frac{\sum GD_t}{\sum N_t} \times 100 \tag{1}$$

where $GD_t$ is the number of trees showing GD in their tree-ring record in year $t$, and $N_t$ is the number of sampled trees for each landslide alive in year $t$. Due to the limited number of samples and detected GD markers available to identify landslide reactivation years, thresholds of $GD_t \geq 2$ and $I_t \geq 15\%$ were used. Additionally, the reported year of landslide events and year of establishment of shade-intolerant tree species were also used to assist our interpretation of the dendrochronological effects of landslide activity.

The establishment of shade-intolerant species is indicative of the development of large gaps in the canopy at some point in the past, and these gaps were most probably caused by landslides [25,26]. Consequently, the ages of individual younger trees from shade-intolerant species were determined (Figure 4e) based on the number of rings counted in the cores and the number of years required for seedlings to reach coring height estimated using an age–height regression relationship. The age–height regression (age (years) = $0.025 \times$ height (cm), $R^2 = 0.27$) was established from 15 specimens of *Ah* (<2.5 m in height) sampled at the Shirakami Natural Science Park of Hirosaki University, 4 km from the study area, where the growth conditions are similar to those in our study area because of their similar elevations. However, the ages for trees of shade-tolerant species and intermediate shade-tolerant species to reach coring height were not estimated, and these trees were used only to identify GD markers on tree-ring width series, as described above. The tree-ring record of these samples were inspected between 1950 and 2019.

## 4. Results and Discussion

### 4.1. Spatial Distribution of Tree Ages and GD in Tree-Ring Width Series

The age of the trees sampled around the N-Ohkawa and Ohkawa landslides was $48.2 \pm 22.4$ years (average $\pm$ 1 SD), with a median of 56 years. The youngest tree was 6 years old and the oldest was 101 years old. Figure 5a shows the spatial distribution of the tree ages of 60 trees used for landslide dating. Older ages tend to be concentrated near the scarps, where the majority of trees were 51–70 years old. Trees of 51–90 years in age were also sparsely distributed on the displaced block and many of these were back-tilted, which suggests that the trees were moved down hillsides during landslide transport by rotation along a circular feature of the sliding surface [11,27]. The younger trees (<20 years old) on the scarps and displaced blocks were the shade-intolerant species *Ah*, *Sb*, and *Ss*. GD markers were not detected in these younger trees (Figure 5).

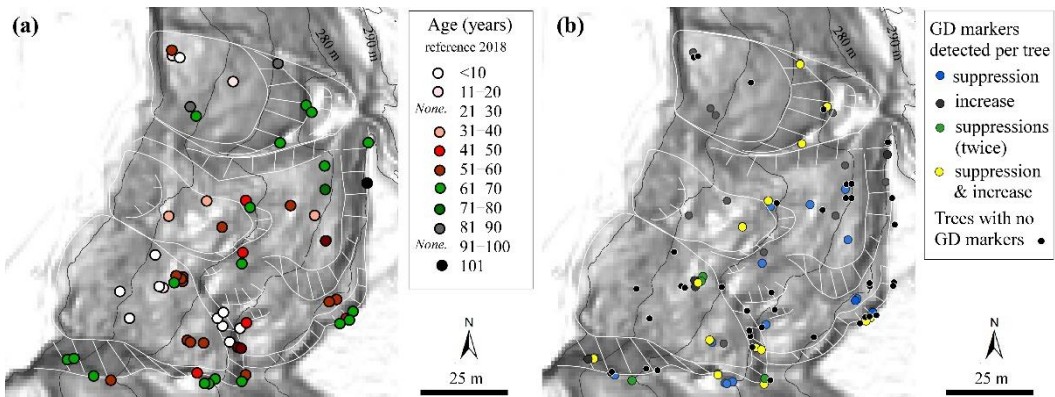

**Figure 5.** Spatial distribution of tree ages and detected GD markers and trees with no GD markers. (**a**) Spatial distribution of the ages of 60 trees sampled for dendrogeomorphological analysis. (**b**) Spatial distribution of detected GD markers for individual trees.

In total, 64 GD events (including 39 moderate GDs) were identified from 47 trees (Figure 5b). Growth suppression (34 GDs, 53%) occurred in slightly more trees than growth increase (30 GDs, 47%). This higher frequency of growth suppression has also been reported in other similar works [14,28]. The highest frequency (45%) of first-detected GD within the tree-ring width series occurred for trees aged between 16 and 30 years. Individual trees with two GDs (e.g., growth suppression and increase or multiple growth-suppression events) were detected mainly on the landslide scarps. Nevertheless, in a few cases, two GD markers were also detected in trees on the landslide blocks.

*4.2. Summary of GD in Tree-Ring Width Series*

The GD markers associated with the N-Ohkawa landslide occurred mainly between 1998 and 2001 (Figure 6a). Samples *Nu.* 9 and 20 on the landslide scarp and the displaced landslide block, respectively, showed wider annual rings in 1998, one year before the landslide event that eyewitnesses reported as occurring in 1999. Two samples (*Nu.* 17 and 18) on the displaced landslide block showed wider annual rings in 1999 in response to the landslide occurrence. Following the event in 1999, trees on the landslide scarp presented wider annual rings in 2000 and 2001. Furthermore, narrow annual rings in 1970, 1982, and 1983 were detected in trees on the landslide scarp. In addition, shade-intolerant trees on the displaced block appeared in the early 2000s, which is a lag of 3–5 years after the growing season following the landslide in 1999 (Figure 6a).

Figure 6b–d summarizes the GD for the Ohkawa landslide. In the eastern area of the landslide, a sample (*Nu.* 82) from the landslide scarp showed wider annual rings in 1956 as the earliest GD marker in the study area (Figure 6b). The majority of GD events appear to be clustered after the 1980s, with six GDs on the landslide scarp and eight GDs on the displaced block. In the northern section of the landslide, samples Nu. 95 and 96 on the landslide scarp showed wider annual rings in 1973 and 1995, respectively. In addition, three trees on the landslide scarp recorded GD markers in 2000 and narrow annual rings were also identified for the same year in a sample (*Nu.* 78) from the displaced block (Figure 6c). Furthermore, wider annual rings were detected in 2005 and 2006 in samples from the displaced blocks. In the southern section of the landslide, 29 GDs were identified between the 1960s and the 2010s (Figure 6d). GDs appear to be concentrated in the 1980s and the late 2000s. In particular, GDs detected between 2006 and 2009 are considered to be the consequence of the landslide event that occurred (based on eye-witness reports) in 2006. Notably, shade-intolerant species appeared for the first time on the scarp and displaced block between the late 2000s and the 2010s, a lag of 2–7 years behind the growing season following the landslide event in 2006 (Figure 6d). The two major eye-witnessed landslide events, the N-Ohkawa landslide in 1999 and the southern section of the Ohkawa landslide in 2006, can be identified in the tree-ring records, which suggests that other GDs detected in the study area may also be indicative of historical landslides that were large enough to remove and damage the trees.

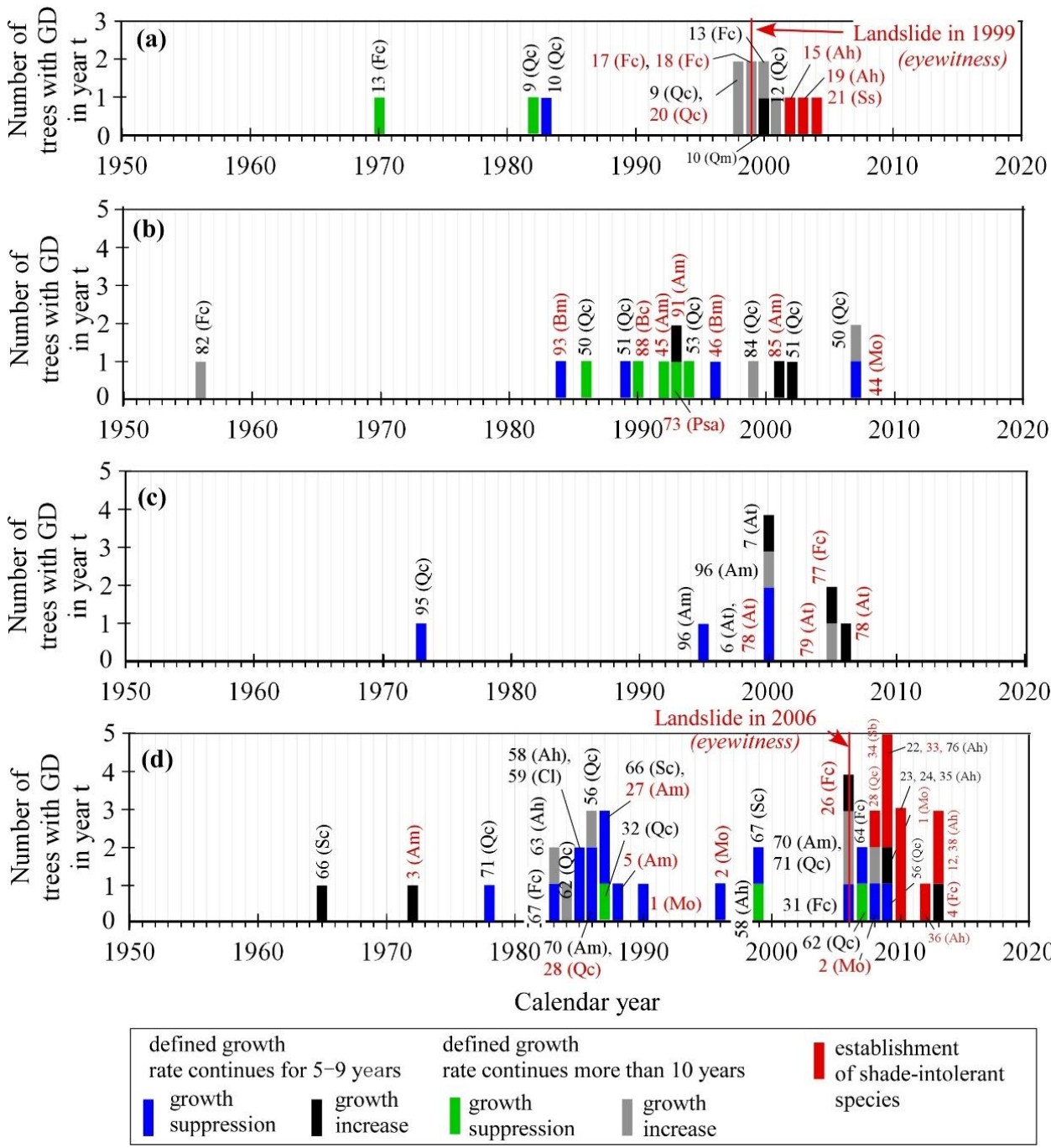

**Figure 6.** Summary of GD markers identified in the trees and year of establishment of shade-intolerant species for the: (**a**) N-Ohkawa landslide, (**b**) eastern section of the Ohkawa landslide, (**c**) northern section of the Ohkawa landslide, and (**d**) southern section of the Ohkawa landslide. Black and red text indicates samples from the landslide scarp and the displaced landslide block, respectively. Sample locations are shown in Figure 3.

*4.3. Dendrochronological Investigations of Spatial and Temporal Patterns of Landslide Reactivation*

The analysis of GD markers enabled the identification of landslide events on the studied slopes (Figure 7). These previous slides are summarized in Figure 8 with reference to the locations of trees with GD markers and field observations. For the N-Ohkawa landslide, apart from the landslide reported in 1999, additional landslide activity was detected in 1998 (Figure 7a). In addition, an event took place in 2000 that was detected using

samples from the landslide scarp, implying an enlargement of the scarp (Figures 7a and 8). In the eastern section of the Ohkawa landslide, for which there are no reported landslides, we detected two landslide events that took place in 1993 and 2007 (Figure 7b). The landslide events in 1993 and 2007 may suggest an episode of regressive enlargement of the landslide scarp along the terrace scarp (Figure 8). This is supported by field observations showing terrain below the landslide scarp with collapsed debris deposited on a pre-existing landslide mass. The observations suggest that the present landslide unit may have grown from gradual accumulation of landslide debris from repeated landslides, in combination with retrogressive enlargement. For the northern section of the Ohkawa landslide, two landslide events were identified in 2000 and 2005 (Figure 7c). These landslide events have not been previously reported; however, the landslide aftermath can be observed on the aerial photograph from 2015 (Figure 1c). Our analysis suggests that a large landslide might have been initiated in 2000 (as three of the four GDs were identified on the scarp; Figure 6c) and experienced further downward movement in 2005 (as GDs were detected on the landslide block; Figures 6c and 8). Furthermore, ongoing movement is evident on the downslope section, which is bounded by a secondary scarp up to 7 m in height in the lower section. This section is cut by a minor gully, in which surface water is concentrated, and which affected the area before and after a local failure in 2017 (Figure 9a). For the southern section of the Ohkawa landslide, apart from the landslide in 2006, two additional previously unknown events were dated to 1986 and 1987 (Figure 7d). Tension cracks were observed on the crown of the Ohkawa landslide along a ridgeline (Figure 9b,c), from which a crack developed into a lateral scarp of the southern section of the Ohkawa landslide in 2006 (Figure 9c).These observations suggest progressive movement prior to the catastrophic failure in 2006 (Figure 8). In addition, in the middle portion of the southern section of the Ohkawa landslide, a disrupted slide (5 m wide, 25 m long, and 20 m travel distance) was also observed in 2009 [29] (Figure 9d). At the northeastern end of the disrupted slide, within the landslide block of the southern section of the Ohkawa landslide, downward slope movement of about 8 m occurred between 2009 and 2014 [29]. The foot of the downslope section is undergoing river toe erosion, this may steepen the slope and facilitate further movement [29,30]. The landslide activity in 1998, as indicated by the GD markers, might also have progressed to become the major event in 1999 on the N-Ohkawa landslide block.

Our dendrochronological study using 60 deciduous broadleaved trees from 12 species for landslide analysis is unique on global scale [12]. In this contribution, we illustrate that the obtained chronology of landslide activity is in agreement with eyewitness reports of the major landslide events in 1999 and 2006, which suggests the GD markers and index values (where $GD_t \geq 2$ and $I_t \geq 15\%$ are adjusted based on the number of disturbed trees available for analysis) employed in this study may provide a critical assessment of past landslide occurrence in the study area and in those areas with similar environmental conditions. Shade-intolerant tree species are typically established between 2–7 years after landslides. However, this lag may reflect the severe erosion that can continue for several years after a landslide, thus limiting tree establishment [26].

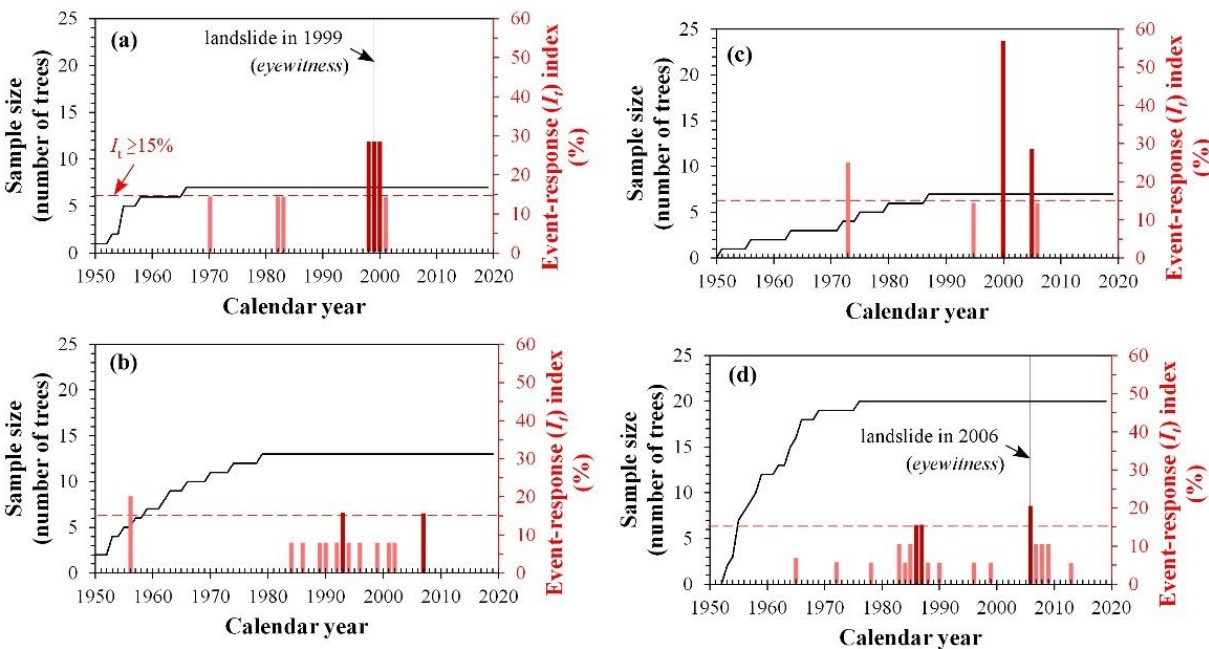

**Figure 7.** Dendrochronological investigations of past landslide events (dark red columns) expressed using the $I_t$ index and number of disturbed trees. (**a**) Chronology of the N-Ohkawa landslide. (**b**) Chronology of the eastern section of the Ohkawa landslide. (**c**) Chronology of the northern section of the Ohkawa landslide. (**d**) Chronology of the southern section of the Ohkawa landslide.

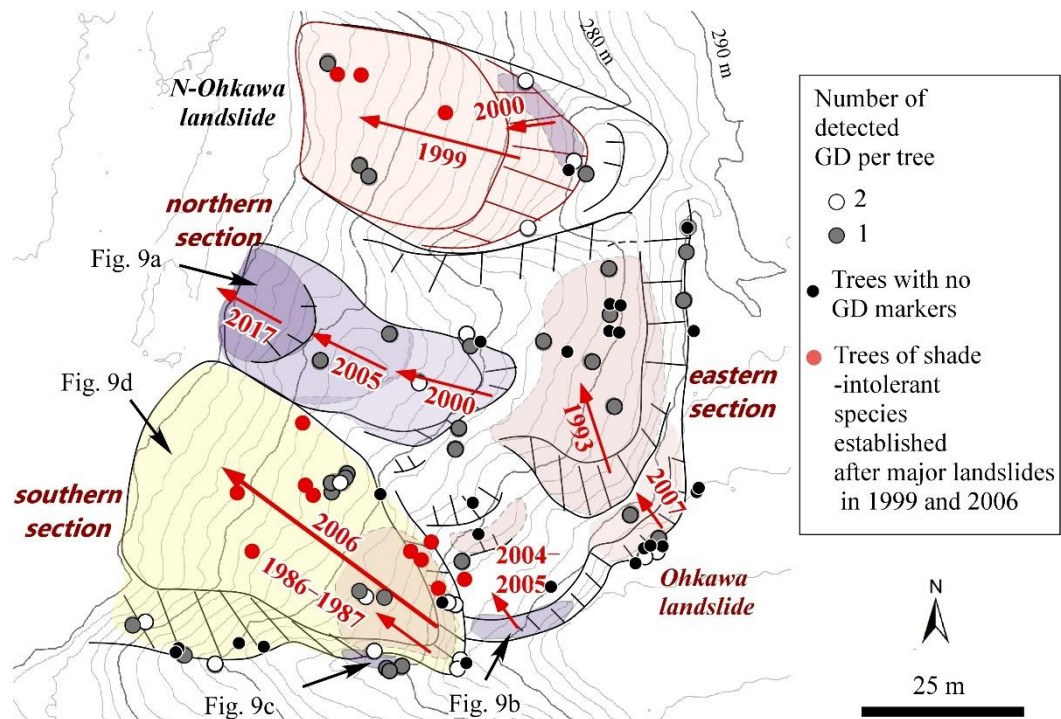

**Figure 8.** Summary of past landslide events in the study area and distribution of detected GDs and trees established after the landslide events.

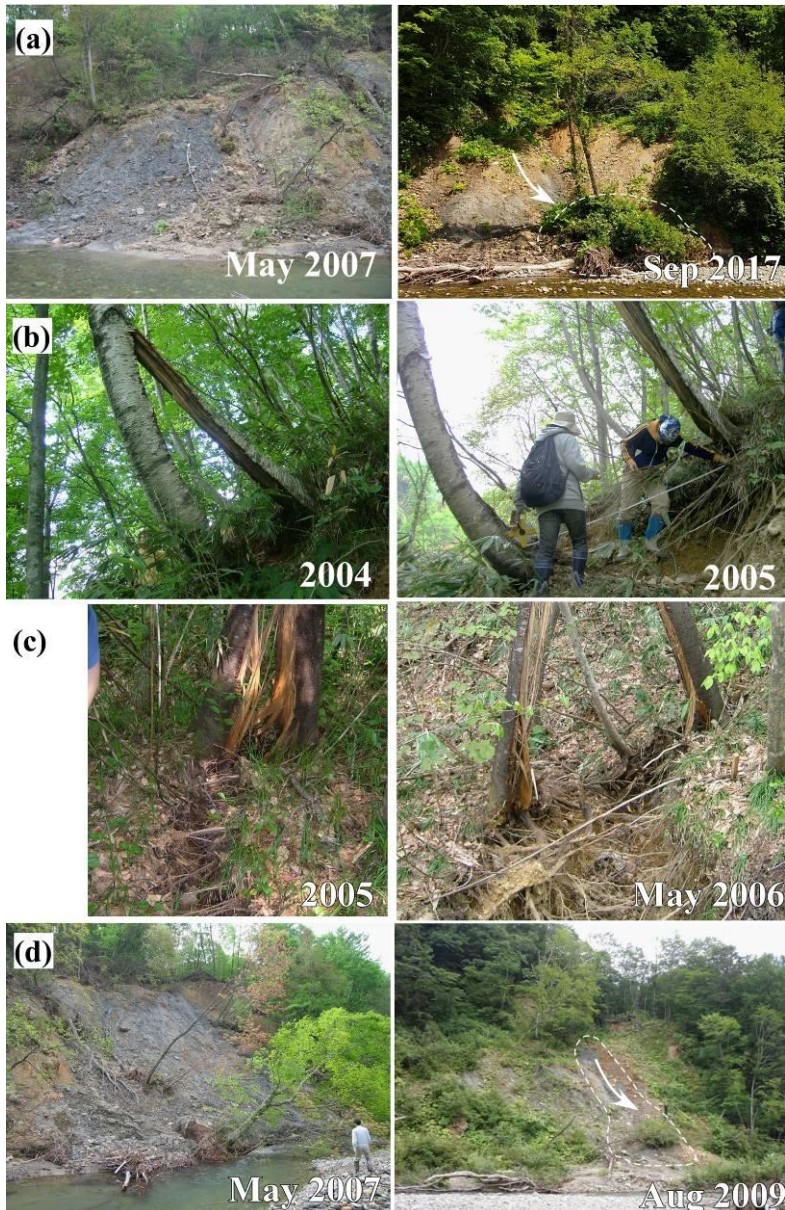

**Figure 9.** Representative examples of slope movements in the study area. (**a**) Ongoing movement on the downslope part of the northern section of the Ohkawa landslide. (**b**) Enlargement of about 2 m tension crack identified in the landslide scarp of the eastern section of the Ohkawa landslide. (**c**) Close-up view of the tension crack. The crack became the lateral scarp of the southern section of the Ohkawa landslide and had enlarged to about 20 m by July 2006. (**d**) A disrupted slide in the downslope part of the southern section of the Ohkawa landslide. The month and year in which photographs were taken are indicated at bottom-right in each panel. Locations of the photographs are indicated in Figure 8.

## 5. Conclusions

The spatial and temporal development of the coherent N-Ohkawa and Ohkawa (consisting of the eastern, northern, and southern sections) landslides were investigated using tree-ring chronologies from tilted deciduous broadleaved trees in the Shirakami Mountains, northern Honshu Island, Japan. In total, we identified 64 GD markers (i.e., periods of growth suppression or growth increase) from 47 trees, as well as the year of establishment of 13 trees from shade-intolerant species over about 70 years.

Our dendrogeomorphological analysis allowed us to identify the GD markers related to two major eye-witnessed landslide events; i.e., the N-Ohkawa landslide in 1999 and the southern section of the Ohkawa landslide in 2006. Shade-intolerant tree species became established after a lag of 2–7 years after the events in response to canopy opening by the landslides. Other GDs were used to reconstruct previously unknown events within the local landslide chronology. The reconstruction of the N-Ohkawa landslide added precursory landslide activity in 1998 and a local enlargement of the landslide scarp in 2000. In addition, the reconstruction of the Ohkawa landslide indicated episodes of regressive enlargement of the landslide scarp from 1993 to 2007 in the eastern section. In the northern section of this landslide, the landslide slope might have been undergoing sliding to form the current landslide scarp observed in 2000. The slope may have moved progressively downwards in 2005 and its secondary scarp on the downslope locally expanded in 2017. In addition, the reconstruction of the southern section of the Ohkawa landslide suggested that progressive movements may have developed in 1986 and 1987; i.e., before the landslide event in 2006.

**Author Contributions:** K.N. conducted field investigations, collected tree samples, and analyzed the tree cores, and also collaborated with the corresponding author in the preparation of the manuscript. C.-Y.T. conducted field investigations, landslide interpretation, and the collection and interpretation of tree samples, and also drafted this manuscript. Y.I. conducted field investigations, collected tree samples, and analyzed and interpreted the tree cores. D.H. conducted field investigations and landslide interpretation. C.-Y.W. conducted landslide interpretation and performed calculations. All authors have read and agreed to the published version of the manuscript.

**Funding:** This work was supported by JSPS KAKENHI Grant Numbers 16K20893 and 19K15257.

**Institutional Review Board Statement:** Not applicable.

**Informed Consent Statement:** Not applicable.

**Acknowledgments:** The authors are grateful to Hajime Makita and Mitsuharu Kudo of Shirakami-Matagisha for providing useful and informative discussions regarding our study. We also thank Hisako Furukawa and Ryunosei Sato of Hirosaki University for their assistance in the field. We acknowledge the Tsugaru Forest Management Office, Tohoku Forest Management Bureau, and Ministry of Agriculture Forestry and Fisheries, Japan for permission to access mountain areas within their territories, and for generous logistical support. The Geospatial Information Authority of Japan provided the LiDAR DEM. We are grateful to three anonymous reviewers whose comments improved the paper.

**Conflicts of Interest:** The authors declare that they have no competing interests.

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
