# Peer review of "Tree-Ring Based Chronology of Landslides in the Shirakami Mountains, Japan"

_water, doi:10.3390/w13091185_

Round 1

Reviewer 1 Report

This is the review of the manuscript from Kinuko Noguchi et al. considered for publication in Water. Authors use dendrogeomorphological approaches to date past landslide events on landslide bodies in northern Honshu Island, Japan. By analyzing abrupt growth changes and dates of seedling establishment they confirmed known dates of landslide events as well as identified previously unknown events.

Major comments:

L. 184-187+Figure 9: Were the trees with no occurrence of GD (33% of trees) considered for this spatial interpolation of recurence intervals? Please, clarify this information. Those trees have a landslide recurrence interval equal to infinity, how was this treated? Trees without GD occurrence should not be removed from the interpolation because they hold information that some area of the landslide was not affected by the movement in the period of the tree lifespan. Moreover, because most of the trees experienced just one or two GD during their lifespan, the interpolated recurrence interval only reflects their age (age distribution across the landslide body) and does not provide true unbiased information about the spatial distribution of movement frequencies. I suggest dropping this analysis to make a manuscript more concise and to reduce the number of figures.

L. 178+Figure 11: The oldest events dated at the eastern and northern sections (1956+1973) are based on only one tree with the occurrence of GD. The estimated It exceeds the threshold of 15% only because there was an extremely small number of living trees in a particular year (less than 7). A single tree with GD gives not sufficient evidence for landslide dating. I suggest including an additional rule to determine landslide events: the event will be dated only if It>0.15 and, at the same time, there will be at least two trees with GD occurrence.

The manuscript contains a very high number of figures (12). This needs to be reduced either by dropping some of them or moving them to supplementary. Some maps might be merged together without reducing (or only slightly reducing) the ammount of information.

Minor comments:

L. 26 Change ‚the caused‘ into ‚causing‘

L. 50: In addition to root damage, the growth suppression can be triggered by additional forms of mechanical damage, e.g. damage to the crown or stems

L. 58-59: According to reviews and metaanalyses of the current state of the art of dendrochronological studies (see e.g. DOI 10.2478/s13386-012-0021-5), broadleaves are frequently used for the dating of landslides compared to other types of mass movements. Please, reformulate.

L. 133: Why were the cores extracted from the upper side of the stem? The standard approach used in dendrogeomorphology is to use a perpendicularly extracted core for the identification of abrupt growth changes. The growth trends in the upper part of the stem might be masked by reaction wood formation in the case of tilted broadleaves. Please, explain your motivation to sample from the upper part of the stem.

L. 167-168: This is not precise. According to figure 4c, the values of GD do not need to be double/half of the previous moving window during consecutive 10 days. You require them just to be continuously above/below 1, right? Reformulate to make the definition of your index clear.

Figure 5: Do not combine colors and shapes in the legend. Use a single symbol (circle) and use a color scale from light to dark colors to indicate the age. This way you can increase the comprehensiveness of the map.

Figure 6: Show also the distribution of trees without GD. The fact, that the tree was not damaged during the entire lifespan is also very important for the assessment of the landslide history and spatial distribution.

L. 279: typo

L. 313-314: Do you mean bottom-right corner?

Reviewer 2 Report

Notes on the text

Line

Note

44-45

How were non-landslide disturbances determined or excluded in the current manuscript?

54

>5% or 10% of what? Width of the previous ring? Frequency of occurrence in the population? I don’t know what is being referred to here.

133

Any particular reason other than convenience to collect cores from the upper-slope side of the stem? Most guides suggest coring parallel to the topographic contour or from the “side-slope”, particularly if the goal is to look for changes in stem orientation relative to the groundline. If cores were collected from the upper side of leaning broadleaved trees, were there indications of tension wood in the cores?

151-152

I realize that visual cross-dating is not particularly quantitative, but can the reader get a sense of how readily or strongly these cross-dated? Other members of those genera are pretty complacent and can be hard to show a common growth signal. I’d put that in Methods but could legitimately go into Results.

163-164

Wouldn’t the five-year moving average (MA) blunt the detection of abrupt change in radial growth? I think using the moving average may well be legitimate, I’m just questioning the use of the term “abrupt”. Is the MA calculated with year t at the midpoint? This introduces the concept that ring width at year t is pre-conditioned by ring width at years t+1 and t+2. Should the reader find that concerning? I suppose my real question is how robust these findings are or are they limited to using this particular technique. I’m thinking of the boundary-line growth pattern approach that Marc Abrams and students have applied e.g., Black and Abrams, 2004. Dendrochronologia or even more generally in Nowacki and Abrams, 1997. Ecological Monographs.

181-182

15% of what? 15% of trees showing the GD at that location or?

This is interesting work! Keep going!

Reviewer 3 Report

The manuscript presented by Noguchi et al., Is well written, is very refined in text and figures. The content and the results obtained are interesting and and provide new insights into the research field. From my point of view, I find the lack of a paragraph dedicated to the dendrochronological method in the methods: I suggest to add in the specific the crossdating and detrending method, that are crucial for a dendrochronological study. I also suggest to emphasize the importance of the research carried out: it is well described what has been done, but it is not stressed enough why it has been done. In my opinion these suggested additions could improve the article.

Round 2

Reviewer 1 Report

The authors responded to my comments correctly and the manuscript shows sufficient quality to be published in Water.

During the reading of the new version of the manuscript, I noticed an inconsistency in L. 127-129 (Figure 2 caption). According to maps, profile I-I' goes through southern Ohkawa landslide body, and profile III-III' goes through N-Ohkawa landslide body. However, the figure caption suggests the opposite. Please, check and possibly correct during the proofs stage.

Reviewer 2 Report

The authors have sufficiently answered my earlier questions. Congratulations!